

# taxalogue: a toolkit to create comprehensive CO1 reference databases

Niklas W. Noll, Christoph Scherber and Livia Schäffler

Centre for Biodiversity Monitoring and Conservation Science, Leibniz Institute for the Analysis of
Biodiversity Change, Bonn, North Rhine-Westphalia, Germany

## ABSTRACT

**Background**. Taxonomic identification through DNA barcodes gained considerable traction through the invention of next-generation sequencing and DNA metabarcoding. Metabarcoding allows for the simultaneous identification of thousands of organisms from bulk samples with high taxonomic resolution. However, reliable identifications can only be achieved with comprehensive and curated reference databases. Therefore, custom reference databases are often created to meet the needs of specific research questions. Due to taxonomic inconsistencies, formatting issues, and technical difficulties, building a custom reference database requires tremendous effort. Here, we present *taxalogue*, an easy-to-use software for creating comprehensive and customized reference databases that provide clean and taxonomically harmonized records. In combination with extensive geographical filtering options, *taxalogue* opens up new possibilities for generating and testing evolutionary hypotheses.

**Methods**. *taxalogue* collects DNA sequences from several online sources and combines them into a reference database. Taxonomic incongruencies between the different data sources can be harmonized according to available taxonomies. Dereplication and various filtering options are available regarding sequence quality or metadata information. *taxalogue* is implemented in the open-source Ruby programming language, and the source code is available at https://github.com/nwnoll/taxalogue. We benchmark four reference databases by sequence identity against eight queries from different localities and trapping devices. Subsamples from each reference database were used to compare how well another one is covered.

**Results**. *taxalogue* produces reference databases with the best coverage at high identities for most tested queries, enabling more accurate, reliable predictions with higher certainty than the other benchmarked reference databases. Additionally, the performance of *taxalogue* is more consistent while providing good coverage for a variety of habitats, regions, and sampling methods. *taxalogue* simplifies the creation of reference databases and makes the process reproducible and transparent. Multiple available output formats for commonly used downstream applications facilitate the easy adoption of *taxalogue* in many different software pipelines. The resulting reference databases improve the taxonomic classification accuracy through high coverage of the query sequences at high identities.

Corresponding author
Niklas W. Noll, N.Noll@leibniz-lib.de

## INTRODUCTION

Great effort is currently being taken to arrive at a comprehensive DNA barcode reference database for all life on Earth (*Hobern & Hebert, 2019*), which has also been fundamental for the mission of the International Barcode of Life Consortium (*International Barcode of Life, 2022*): saving the living planet and cataloging all multicellular species before the first half of the century. DNA barcodes are short marker-gene sequences that are ideally conserved at species level with sufficient genetic differentiation to distinguish even closely related sister taxa (*Hebert et al., 2003*; *Hebert, Ratnasingham & De Waard, 2003*). Many different barcode markers are used for different taxa, but the most often used animal barcode is the Folmer region (*Folmer et al., 1994*) of the mitochondrial cytochrome c oxidase subunit I (CO1) gene, which is part of the respiratory complex and is known to have, in general, a high resolution until species level (*e.g.*, *Hebert et al., 2003*; *Hebert, Ratnasingham & De Waard, 2003*; *Fišer Pečnikar & Buzan, 2014*; *Huemer et al., 2014*). To identify specimens even without taxonomic expertise, the same barcode region from unknown organisms is sequenced and compared to barcode sequences of already identified specimens stored in a reference database. New sequences can be compared directly with an online source database using identification services such as those provided by GenBank (*Sayers et al., 2022*), the Barcode of Life Data System (BOLD; *Ratnasingham & Hebert, 2007*), or the German Barcode Of Life Initiative (GBOL; *Geiger et al., 2016a*). Since large online databases are subject to constant changes (*e.g.*, *Porter & Hajibabaei, 2018a*; *Porter & Hajibabaei, 2018b*; *Sayers et al., 2022*), self-created reference databases are often used instead (*Robeson et al., 2021*); they require more work and expertise but provide complete control over the sequences and make taxonomic identification reproducible (*Robeson et al., 2021*). Given the large number of sequences generated by metabarcoding, where the DNA from many organisms is simultaneously sequenced (*Taberlet et al., 2012*), a self-created reference database can also speed up the identification process (*Macher, Macher & Leese, 2017*).

The primary goal of a DNA barcode reference database is to provide taxon names for sequences. Taxon names are like other carefully circumscribed abstractions: good names subsume ecological observations and evolutionary theories (*Franz, 2005*). Therefore, scientific species names are a link to the accumulated knowledge of a species in time (*Grimaldi & Engel, 2005*) and much of biology relies on them (*Agnarsson & Kuntner, 2007*). However, synonyms, taxonomic disagreements, and revisions have received little attention in using DNA barcode reference databases (*Leray et al., 2019*; *Pappalardo et al., 2021*; *Piper et al., 2021*). Their effects on the interpretation of metabarcoding results remain unexplored, even though proper taxonomic name usage is a prerequisite for any reliable conclusion (*e.g.*, *Bortolus, 2008*). Taxa lists derived from metabarcoding results depend on the composition of the used reference database: taxon names in the reference database might be based on a particular taxonomic opinion, used identification literature, prior taxonomic harmonization (*Ratnasingham & Hebert, 2007*; *Schoch et al., 2020*), reverse taxonomy (identification by its sequence and not morphology) (*Weigand et al., 2019*), and more. Even an accepted name in a source database could convey distinct taxonomic concepts

(*Berendsohn & Geoffroy, 2007*). Since taxa are potentially described based on different paradigms, taxonomists might prioritize different traits (*Thompson, 1993*). Consequently, taxonomists might apply the same name, although they have a distinct definition of that taxon (*Kennedy, Kukla & Paterson, 2005*). The meaning of a name is unclear without mentioning the taxonomic circumscription on which an identifier based the specimen identification (*Berendsohn, 1995*). Harmonizing taxon names is an often-used step to ensure an up-to-date taxonomy and successful data integration from multiple sources (*Grenié et al., 2022*). Since manual harmonization might not be actionable for studies investigating a broad range of taxa, or a diverse taxon such as Arthropoda, an automated approach might be the most obvious. Data aggregators such as the National Center for Biotechnology Information (NCBI; *NCBI, 1988–2023*) or the Global Biodiversity Information Facility (GBIF; *GBIF, 2023*) provide a resolved taxonomy (*Schoch et al., 2020*; *GBIF Secretariat, 2022*) by acting as a decisive authority in the case of taxonomic disagreements and can be used to automatically harmonize data from different sources.

Besides the influences of nomenclature and taxonomy on the source databases, data quality and coverage are also essential for the condition of the used reference database. Comprehensive taxonomic coverage of a reference database is necessary for reliable identifications (*Meyer & Paulay, 2005*; *Vences et al., 2005*; *Ekrem, Willassen & Stur, 2007*). A sufficient sampling of each taxon has been stressed as an initial requirement for DNA Barcoding (*Sperling, 2003*), and its importance continues to be emphasized (*Phillips, Gillis & Hanner, 2019*). For taxa with high intraspecific variation, sampling the whole geographic range might be necessary for appropriate identification (*Lou & Golding, 2012*; *Geiger et al., 2016b*). However, the observed genetic differentiation between closely related taxa might also decrease with an increase in the geographic scale of the reference database, impairing the identification process. Therefore, regional reference databases have been suggested (*Bergsten et al., 2012*). Despite significant efforts to complete these reference databases, commonly used sources such as GenBank (*Sayers et al., 2022*) and BOLD (*Ratnasingham & Hebert, 2007*) still have exclusive CO1 records (*Porter et al., 2014*; *Macher, Macher & Leese, 2017*; *Curry et al., 2018*; *Porter & Hajibabaei, 2018a*; *Pentinsaari et al., 2020*; *O'Rourke et al., 2020*; *Porter & Hajibabaei, 2020*; *Robeson et al., 2021*; *Nakazato & Jinbo, 2022*) and coverage is reduced when using just one source. Filtering may become necessary when data quality in reference databases is insufficient (*Meyer & Paulay, 2005*; *Nilsson et al., 2006*; *Collins & Cruickshank, 2013*).

The aforementioned issues and circumstances clarify that care is required when creating a reference database. Several software solutions have been developed to create custom reference databases (*Macher, Macher & Leese, 2017*; *Bengtsson-Palme et al., 2018*; *Palmer et al., 2018*; *Richardson et al., 2018*; *Heller et al., 2018*; *Keller et al., 2020*; *Arranz et al., 2020*; *Robeson et al., 2021*; *Piper et al., 2021*; *Meglécz, 2023*; *Keck & Altermatt, 2022*) or to provide ready-to-use reference databases (*Leray et al., 2018*; *Porter & Hajibabaei, 2018b*; *O'Rourke et al., 2020*; *Leray, Knowlton & Machida, 2022*; *Magoga et al., 2022*). However, only some can integrate multiple CO1 database sources (*Macher, Macher & Leese, 2017*; *Bengtsson-Palme et al., 2018*; *Porter & Hajibabaei, 2018a*; *Arranz et al., 2020*; *Piper et al., 2021*; *Meglécz, 2023*; *Keck & Altermatt, 2022*). To the best of our knowledge, no software

currently available allows the exploration of distinct taxonomic harmonization strategies while also including data from GBOL, having extensive sequence filtering options, creating reference databases with different geographical scales (countries, continents, biogeographic realms, or user-defined ArcGIS shape files), dereplication, and providing multiple ready-to-use outputs for common downstream analysis applications. To close this gap, we developed *taxalogue* (https://github.com/nwnoll/taxalogue). In this paper, we demonstrate the suitability of this toolkit to create comprehensive and customized reference databases and compare them with already available CO1 reference databases for arthropods.

## MATERIALS & METHODS

The current version of *taxalogue* can create reference databases of the CO1 Folmer region (*Folmer et al., 1994*) for animals. CO1 sequences from animal specimens are referred to as "sequences" or "records" in the following. We envisage the implementation of additional markers and a broader range of taxa for upcoming releases. See Fig. 1 for an overview of *taxalogue* main functions and consider using *taxalogue* with the "--help" command, or visit the GitHub webpage (https://github.com/nwnoll/taxalogue).

### Implementation
We implemented *taxalogue* with the Ruby programming language. *taxalogue* has been tested on Ubuntu 18.04, 20.04, and 22.04. The source code complies with Ruby $\geq$ 2.6.3+ until version 3.2.2. Future patches will ensure compatibility with new Ruby releases and Ubuntu versions. Data storage and retrieval are set up with a current version of SQLite (*Hipp, 2023*). Additional dependencies are listed in the Gemfile. The source code is licensed under the GNU General Public License v3.0 and available at https://github.com/nwnoll/taxalogue.

### Backbone taxonomy
*taxalogue* automatically downloads backbone taxonomy files and imports them into an SQLite (*Hipp, 2023*) database. *taxalogue* relies on a backbone taxonomy database to check and format taxonomic information from multiple sources. Users can use the "setup" subcommand to reset the taxonomies or import them separately. We optimized the database model for query speed through indexing, which decreases program runtime after the database has been built. However, importing millions of taxonomic records into the database will take some time, depending on the machine used. *taxalogue* provides the option to use the GBIF backbone Taxonomy (*GBIF Secretariat, 2022*), NCBI Taxonomy (*Schoch et al., 2020*) or none. *taxalogue* resolves and imports homonyms based on a list provided by *GBIF Secretariat (2022)*.

### Download
*taxalogue* collects data from up to three different online sources to generate various outputs that users could use as a reference database for taxonomic assignment of DNA sequences. The online sources currently available are BOLD (http://www.boldsystems.org/), NCBI GenBank (http://www.ncbi.nlm.nih.gov/genbank/), and GBOL (https://bolgermany.de/gbol1/ergebnisse/results). The retrieval of sequences and specimen information, such

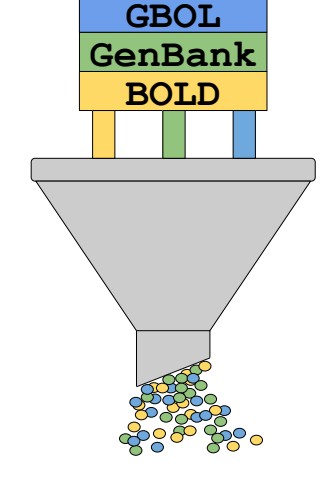

**Download**
from **all databases,**
or pick the ones you
want

**Filter**
by **sequence properties**
(e.g. Ns, length)
by **taxonomic lineage**
(e.g. name, rank)
by **other metadata**
(e.g. location, realm)

**Harmonize**
taxon name to a
**reference taxonomy**
(e.g. NCBI, GBIF) or

**allow synonyms**
according to chosen
reference taxonomy

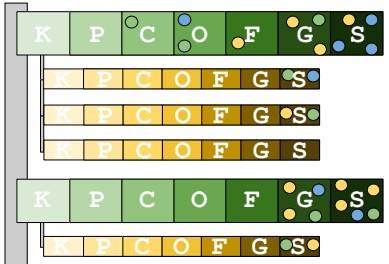

| rank | taxon | sequence | # |
|---|---|---|---|
| species | *A. cerana* | ACCTAG | 1 |
| **species** | ***A. florea*** | **ACCTAG** | **9** |
| family | Apidae | ACCTAG | 5 |

**Dereplicate**
and choose taxon if
the same sequence has
**differing taxon
assignments**
(e.g. LCA, random)

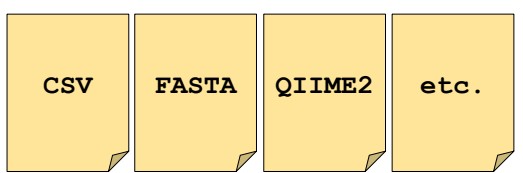

**Output**
generate outputs in
**several formats**
(e.g. QIIME2, FASTA)

**Figure 1** **Overview of main taxalogue functions from the download of records to output generation.**
For more information use taxalogue with "--help". K, kingdom; P, phylum; C, class; O, order; F, family;
G, genus; S, species.

as taxonomic name and locality, varies between the three sources, as explained below.
To prevent unnecessary downloads, *taxalogue* checks if the user has already downloaded
records for a taxon.

NCBI GenBank: Many attempts to download records *via* web queries (*e.g.*,
https://eutils.ncbi.nlm.nih.gov/entrez/eutils/) yielded incomplete downloads, even if we
implemented the recommended waiting times. Therefore, the primary download strategy
used in *taxalogue* is downloading the whole GenBank release for the user-specified

taxon. If, for example, the user wants records of the taxon Arthropoda, *taxalogue* will download all invertebrate records (gbinv*.seq.gz) from the latest GenBank release (https://ftp.ncbi.nlm.nih.gov/genbank/). We implemented waiting times to avoid server overload. If a download fails, *taxalogue* restarts it after an extended waiting period. The download of the current GenBank release ensures the complete retrieval of all records for a particular taxon but has the disadvantage of needing more disk space.

BOLD: There are two approaches to download records from BOLD. We recommend that the user downloads the current data package snapshot from https://boldsystems.org/index.php/datapackages (Data package tar.gz compressed) and uses *taxalogue* with the extracted '.tsv' file (classify --bold_release/path/to/bold_release.tsv; but see README.md). The user must have a boldsystems.org account and be logged in for this approach. The alternative is to use the BOLD API (download --bold), with its drawbacks mentioned below. If the API is used, the user-specified taxon is queried against the public data API (http://www.boldsystems.org/index.php/api_home) for combined data. In general, queries for taxa with many records available, as in Arthropoda, will fail. The taxon for which the download failed will be subdivided into the next lower taxa to circumvent this problem. Supported taxonomic ranks are kingdom, phylum, class, order, family, genus, and species. As with the NCBI GenBank download, we implemented waiting times and retries to avoid overloading the server. We parallelized the download to speed up data retrieval, and the user can specify the number of threads used to retrieve the records. The default value is five threads, and users should make changes with caution. Too many threads could overload the BOLD server and ultimately result in a complete shutdown for the user. Because of this, we recommend not to increase the number of threads used simultaneously to more than the default value of five.

A downside of the taxonomic subdivision into lower ranks is that records determined solely to the taxonomic rank for which the download has failed are unavailable. If, for example, the user specified to download all Arthropoda records, the downloaded results will not include those records that have only Arthropoda as name information. However, it is a benign problem since higher taxonomic ranks (*e.g.*, Arthropoda) would still be covered by lower ranks (*e.g.*, Coleoptera) of that taxon in the subsequent taxonomic assignment step. This is because taxonomic assignment to higher ranks requires less sequence similarity than lower taxonomic ranks. These are rare cases, and records with a greater taxonomic resolution are preferred.

GBOL: The latest GBOL dataset release (bolgermany.de/gbol1/release/GBOL_Dataset_Release-20210128.zip) is provided as a zip file. *taxalogue* will download the file and extract the CSV file. Since the GBOL release has some rank inconsistencies, meaning that not all ranks are used at the same position in the higher classification, *taxalogue* will add those missing ranks. Depending on the user-specified options, this might be necessary to enable merging of all three source databases. The GBOL database is intended as a reference barcode source for Germany. Therefore, it consists mainly of specimens collected in Germany. Since these specimens might also occur in neighboring countries or could be invasive in, for example, North America, it might still be of value to include these records in reference databases for studies from other countries.

## Filtering

The user can filter records by properties such as the number of ambiguous bases (Ns), length, minimal available taxonomic rank, and others. More information is available with the "filter --help" command. It is also possible to only retain records collected in one or multiple countries, continents, or biogeographic realms ("region --help" will provide more information). Since some records have the same sequence, a dereplication step is applied by default. Dereplication removes redundant data and decreases the size of the reference database, which could speed up further downstream analysis. During dereplication, multiple comparisons occur if records have the same sequence but differing taxonomic information. If everything except the taxonomic resolution remained unaltered, the dereplication procedure will favor records with greater taxonomic resolution. The lowest common ancestor is chosen for records with differing taxonomic information at the same rank, given they also have the same number of records. *taxalogue* will choose a record as the correct one if it has more records. Even though we are aware that this is subject to taxonomic bias, it is a pragmatic way to conserve taxonomic resolution; for a reference, see *Leray et al. (2019)*, who investigated clusters with multiple taxon names, and in 95% of cases the most abundant taxon name was labeled as the correct one. *taxalogue* processes the GenBank format and amino acid translation with functions from the Ruby gem "bio" version 2.0.1 (*Goto et al., 2010*).

## Harmonization

Harmonization means that the taxonomy of a record is mapped onto a backbone taxonomy. The taxonomy from the downloaded record is mapped against, for example, the NCBI Taxonomy, and only the standard ranks (kingdom, phylum, class, order, family, genus, and species) will be displayed in the reference database. This action is optional and does not need to be used, although it is the current default setting (to disable harmonization, use the "taxonomy --unmapped" option). It also checks if the taxon of the record is the currently accepted taxon, according to the backbone taxonomy. If the downloaded record has a taxon name considered a synonym, it will replace the name with the accepted name unless the user allows synonyms. This action will be noted and is available in the comparison file. If *taxalogue* could find neither the accepted name nor a synonym, the next higher taxon from the downloaded record is checked against the backbone taxonomy until it finds a match. If it finds a match, it will display the matched higher rank as the actual determination. This action is not without drawbacks and is, therefore, optional. Since some taxonomic classifiers compare the taxon information of each rank, synonyms would be regarded as different taxa and result in a lower bootstrap value, which could lead to the exclusion of some ranks for some sequences.

The already mentioned "taxonomy --unmapped" option does not do any harmonization. It merges the downloads without mapping them onto a backbone taxonomy. This has some consequences, for example: the records from the GBOL Database provide the kingdom name Animalia, whereas the NCBI GenBank records use the name Metazoa, and the BOLD records do not have any kingdom information available. The same taxa with differing taxonomic information on some ranks might affect downstream

analysis. If the user runs *taxalogue* with the "--unmapped" option, users should be aware that different taxonomic classifications within your dataset might occur. A Ruby script "scripts/replace_taxon_name_for_rank.rb" can change taxon names for each rank.

### Name cleaning

Since many names from online sources include digits or terms specifying accuracy and are not part of a valid taxonomic name, some name cleaning will be performed. Digits are not allowed and will be erased from the name. Terms belonging to open nomenclature, like aff., cff. and others were taken from *Matthews (1973)* and will be erased, leaving only the name parts that could be considered valid (*e.g.*, "*Apis* cf. *mellifera*" would result in "*Apis*"). Or in other words: *taxalogue* only uses name parts, where the identifier of that particular specimen has been sure about the correctness of the identification. Also, other name parts as sp. or spp. will be erased. If harmonization is enabled and no representative of this name could be found for this name, the Ruby library biodiversity ($\sim$>5.1, $\geq$ 5.1.2) is used if no backbone taxonomy has been specified to classify the records ("taxonomy --unmapped").

### Output formats

*taxalogue* provides multiple output formats for the reference database. Differing output formats provide distinct information depth. The table format is a tab-separated text file that contains location information. *taxalogue* creates it by default and is required for some optional processing (*e.g.*, "scripts/replace_taxon_name_for_rank.rb" relies on the table file). A fasta file and a comparison file are also created by default. The comparison file shows the accepted names according to a chosen backbone taxonomy and their synonyms. Additionally, output files in the format for dada2, kraken2, qiime2, SINTAX can be generated.

### Case study

To test a reference database created by *taxalogue* against three published CO1 reference databases, we searched metabarcoding publications for OTU sequences or mock communities to use them as queries. The tested reference databases consist of records from different sources and filtering procedures (see Table 1). The used query datasets are shown in Table 2 and were selected to cover different regions of the world and different sampling methods. Any preprocessing and filtering of the databases is described in "ref_db_taxalogue/worklow_ref_db_taxalogue.txt" and in "benchmark/workflow_benchmark.txt" (see *Noll, Scherber & Schäffler, 2023*).

The main method used to compare the reference databases was a top-hit identity distribution (THID; *Edgar, 2018*). A THID shows the distances between a query dataset, *e.g.*, OTU sequences, and a reference database. The number of best hits between a query sequence and a reference database is used herewith as a function of sequence identity. We generated the THIDs with VSEARCH version 2.14.1 (*Rognes et al., 2016*), with the "--usearch_global" (*Edgar, 2010*) command and the essential options "--id 0.7 --maxaccepts 8 --maxrejects 128 --top_hits_only --maxhits 1 --userfields query+target+id". Computed identities were subsequently rounded to integers and summarized with a custom script. We created the figures with Google Drawings, R version 4.1.3 (*R Core Team, 2023*) and the

**Table 1** **Summary of reference databases used in the benchmark.** Database = Arthropoda CO1 reference database name; midori (https://www.reference-midori.info/download.php); porter (https://github.com/terrimporter/CO1Classifier/releases/tag/v4-ref); tidybug (https://doi.org/10.5281/zenodo.3929511); taxalogue (https://doi.org/10.5281/zenodo.6586571), #sequences = total number of sequences, min sequence length = smallest sequence length in reference database, BOLD = download date, GBOL = download date, GenBank = download date, reference = publication reference.

| database | #sequences | Min sequence length | BOLD | GBOL | GenBank | Reference |
|---|---|---|---|---|---|---|
| midori | 2,086,807 | 100 bp | none | none | 2022-02-15 | *Leray et al. (2018)* |
| porter | 888,696 | 500 bp | 2015-12-31[a] | none | 2019-04 | *Porter & Hajibabaei (2018b)* |
| taxalogue | 2,921,104 | 400 bp | 2022-02-02 | 2021-01-28 | 2021-12-15 | This publication |
| tidybug | 1,841,946 | 100 bp | 2019-02-24 | none | none | *O'Rourke et al. (2020)* |

**Notes.**
[a]BOLD data releases from December 31, 2010 till December 31, 2015.

**Table 2** **Summary of the query datasets used in the benchmark.** Country = country of sample; Sampling method = device or method for sampling of specimens; Habitat = natural habitat where sampling did take place.

| Country | Sampling method | Habitat | Taxon | Reference |
|---|---|---|---|---|
| Canada | kick net | benthic zone | Macrozoobenthos | *Porter et al. (2014)* |
| Canada | Malaise trap | Grassland, forested pond | Arthropoda | *Steinke et al. (2021)* |
| China | Malaise trap | Mock[a] | Arthropoda | *Yu et al. (2012)* |
| China | Malaise trap | Mock[a] | Arthropoda | *Yang et al. (2021)* |
| Costa Rica | Malaise trap | Rainforest | Arthropoda | *Porter et al. (2014)* |
| Germany | Malaise trap | Meadow | Arthropoda | *Elbrecht et al. (2021)* |
| Honduras | Canopy fogging | Canopy | Arthropoda | *Creedy, Ng & Vogler (2019)* |
| Portugal | Automatic light traps | Cork oak woodlands | Arthropoda | *Mata et al. (2021)* |

**Notes.**
[a]mock = sampled from multiple locations and potentially different habitats, taxon = expected organism group, reference = publication reference.

R packages dplyr (*Wickham et al., 2023*), ggplot2 (*Wickham, 2016*), ggpubr (*Kassambara, 2023*), ggstance, gridExtra and according dependencies. See the folder "benchmark" in the associated data for complete commands, scripts, and the whole workflow (*Noll, Scherber & Schäffler, 2023*).

Based on the aforementioned THID data, we calculated ranks for all reference database/query combinations at 100% identity (meaning that the query sequence and the most similar reference database sequence had identical nucleotides in the overlap). Ranks ranged from 1 to 4, whereas rank 1 means the fewest best hits at 100% identity and rank 4 the most. We calculated the ranks with the "dense_rank" function from the R package dplyr (*Wickham et al., 2023*). Equal values ("ties") are replaced by their minimum values (*e.g.*, if two reference databases had the most and an equal amount of 100% best hits against all queries, both reference databases would get rank 3 instead of rank 4).

We further investigated the midori, *taxalogue*, and tidybug reference databases: 10 × 5,000 sequences were randomly subsampled for each reference database with the "--fastx_subsample" option of VSEARCH version 2.14.1 (*Rognes et al., 2016*). Each subsample was subsequently used as a query against all reference databases, itself excluded, with the same commands as for the THID generation. We excluded the porter reference database for this benchmark since it only included a subset of BOLD records until the end

of 2015 and was primarily composed of GenBank records. This biases the representation of subsampled sequences. By chance alone, fewer BOLD than GenBank sequences would be sampled as queries; therefore, tidybug, with only BOLD sequences, gets fewer best hits at high identities.

## RESULTS

The top-hit identity distribution (THID) for four reference databases and eight distinct query datasets is shown in Fig. 2. The THIDs show how well a reference database represents a query. Most THIDs show a skew to the left to higher identities, which means that the highest proportion of queries has their best matches to very similar reference database sequences. Therefore, reference databases with more hits at high percent similarities have better coverage of the queries. The reference database created by *taxalogue* shows for most queries the best coverage at high identities and fewer hits with low identity. This is also true for the tidybug and midori reference databases, but here we see more variation with different query datasets (see Fig. 3). The porter reference database reflects, in all cases, the query datasets the least good. However, kick samples from Canada (see Fig. 2C) had a peak at 98% sequence identity, with only a small number of best hits at higher identities. Additionally, Malaise trap samples from China (see Fig. 2E) had a peak around 84% sequence identity and a smaller, second peak at 100% identity for most reference databases.

In Fig. 4, the THIDs of three reference databases are shown with subsampled queries taken from other reference databases than themselves. *taxalogue* had the most hits at 100% identity with sequences from midori or tidybug. Accordingly, the reference databases midori and tidybug had more hits with lower identities, like 99% and 98%. This shows that *taxalogue* provides more exclusive sequences and generally offers better coverage than the other reference databases.

## DISCUSSION

We presented *taxalogue*, a new toolkit to create reproducible reference databases. Using a case study, we showed that *taxalogue* creates reference databases that generally best represent the test cases from multiple areas and trapping devices. *taxalogue* addresses mentioned issues of the current source and reference databases. However, some problems require a major structural change in the source databases (*e.g.*, provision of taxon concepts, data integrity indicators through digital fingerprints, enforced data quality through constraints), and our approach represents only the most appropriate solution under the given circumstances.

### Comprehensive reference databases

Reference databases are the foundation of taxonomic identification *via* metabarcoding, and resulting taxa lists depend on the quality of the underlying reference database. Therefore, creating a reference database should have a high priority. We showed in our case study that combining records from multiple source databases generally leads to a better representation of the test cases. A better representation with higher identities between query and reference

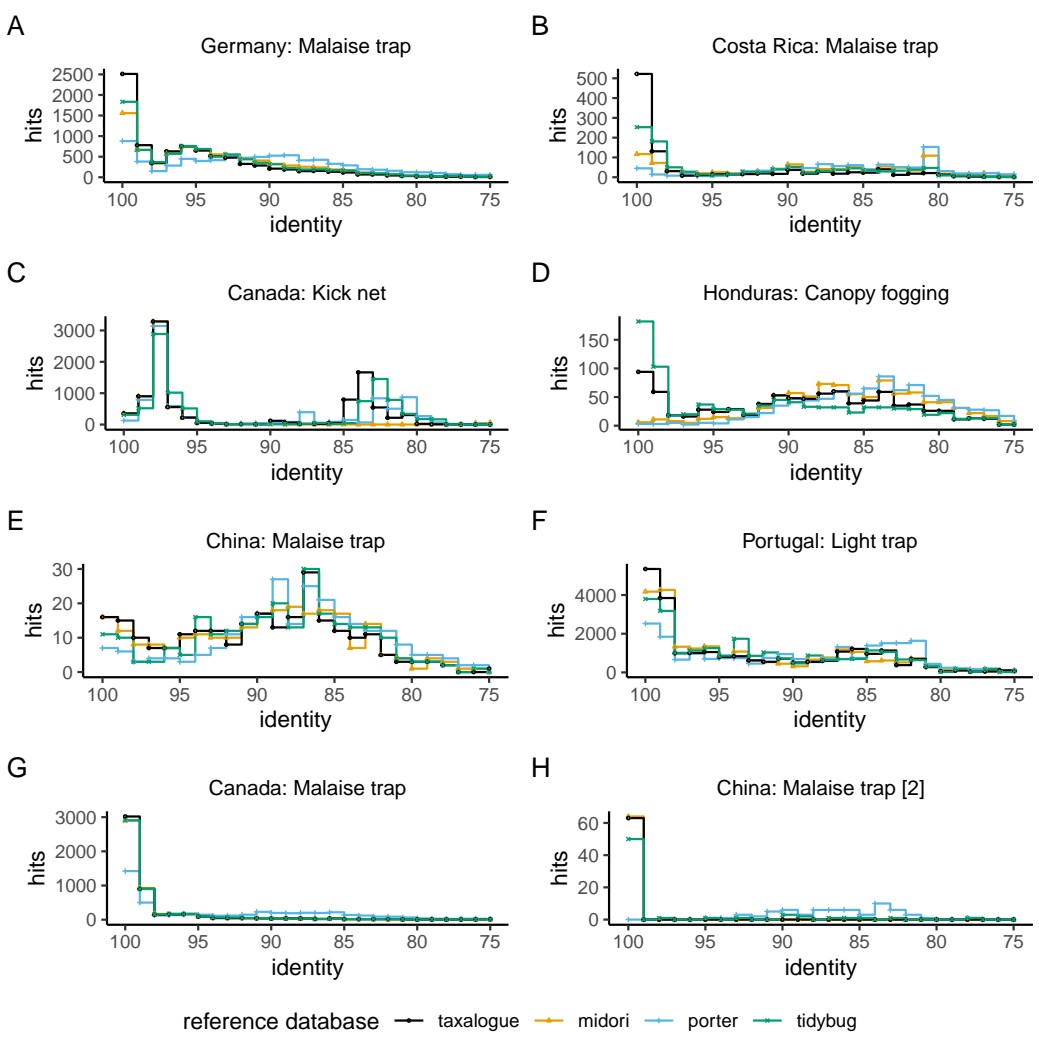

**Figure 2** **Top-hit identity distribution for 4 reference databases and eight queries.** The number of best hits as a function of sequence identity for a selection of CO1 reference databases and query datasets (see Table 1 for reference database descriptions and Table 2 for the query datasets), depicted as a stair step diagram. Identity = percent similarity between a query sequence and its best hit in a reference database. Hits = number of best matches between query and reference database sequences at a certain identity percentage.

database is crucial for correct taxonomic predictions (*Edgar, 2018*). *taxalogue* produces a reference database with the best coverage at high identities for most tested queries, enabling more accurate and reliable predictions with higher certainty than the other reference databases tested. Yet, we cannot conclude that a better representation by sequence identity would result in a more reliable reference database per se. More extensive reference databases have higher coverage, but this may be due to records with incorrect annotations (*Edgar, 2018*). The reference database from the case study created with *taxalogue* consists of records from three source databases. This potentially increases the total amount of erroneous records since several cases of misidentifications have been found in the source

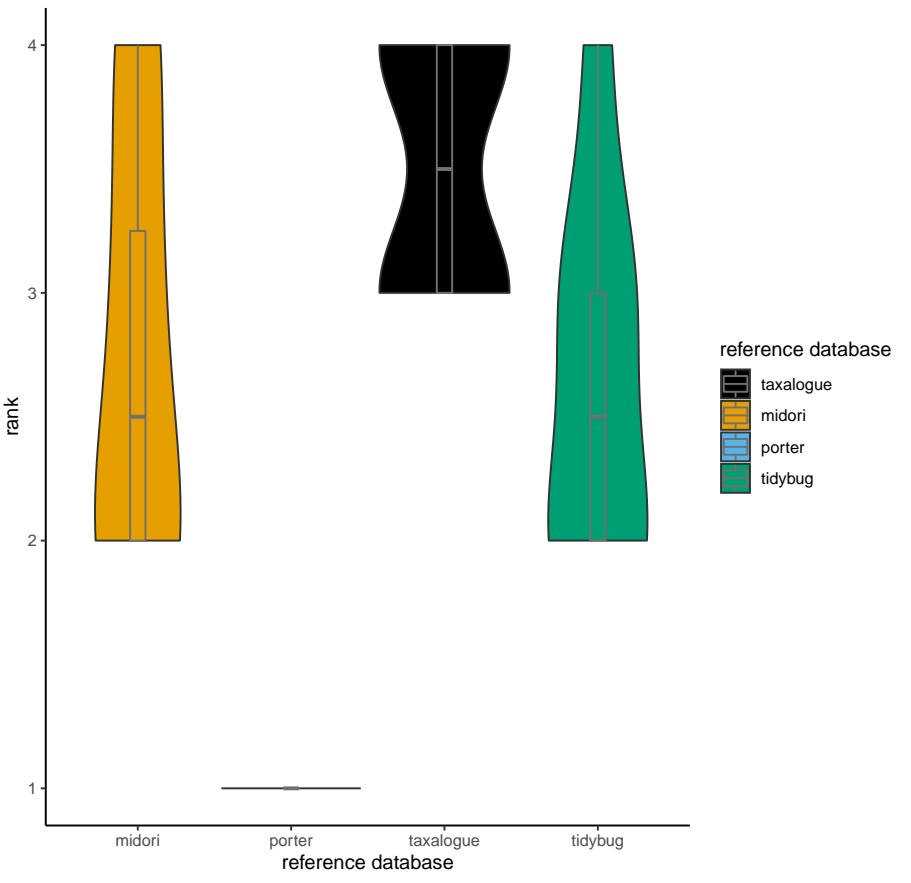

**Figure 3** **Violin plots showing the top-hit identity distributions at 100% identity from all reference database/query combinations.** The width of a violin indicates how often a reference database achieved a rank; the height shows the variation in achieved ranks. Each reference database was tested with 8 queries. rank = Ranks range from 1 to 4, where rank 1 corresponds to the fewest best hits at 100% identity and rank 4 to the highest number of best hits (if reference databases had the same number of best hits, they share the next lower rank), reference database = Arthropoda CO1 reference database name.

databases GenBank and BOLD (*e.g.*, *Meier & Dikow, 2004*; *Becker, Hanner & Steinke, 2011*; *Lis & Lis, 2011*; *Lis, Lis & Ziaja, 2016*; *Jin et al., 2020*; *Radulovici et al., 2021*; *Kjærandsen, 2022*). But since *Leray et al. (2019)* found a surprisingly low error rate (0.44% to 2.56%) in GenBank for Arthropoda CO1 sequences at the genus level and similar results were found at the species level in a study that investigated both GenBank and BOLD (*Jin et al., 2020*), the baseline of expected errors should be low. Furthermore, the also included GBOL source database has more strict quality standards, and only records from species experts are accepted (*Coleman & Radulovici, 2020*); even though this has not been empirically tested, we would expect a similar error rate for records from GBOL. Although the general trend of source database quality points in a positive direction, the methodology of the aforementioned studies prevents a conclusion in this regard. *Leray et al. (2019)* did not investigate incongruities of species names. Due to increased difficulty in assigning species names (*e.g.*, *Sweeney et al., 2011*; *Ko et al., 2013*), we expect a higher error proportion at the

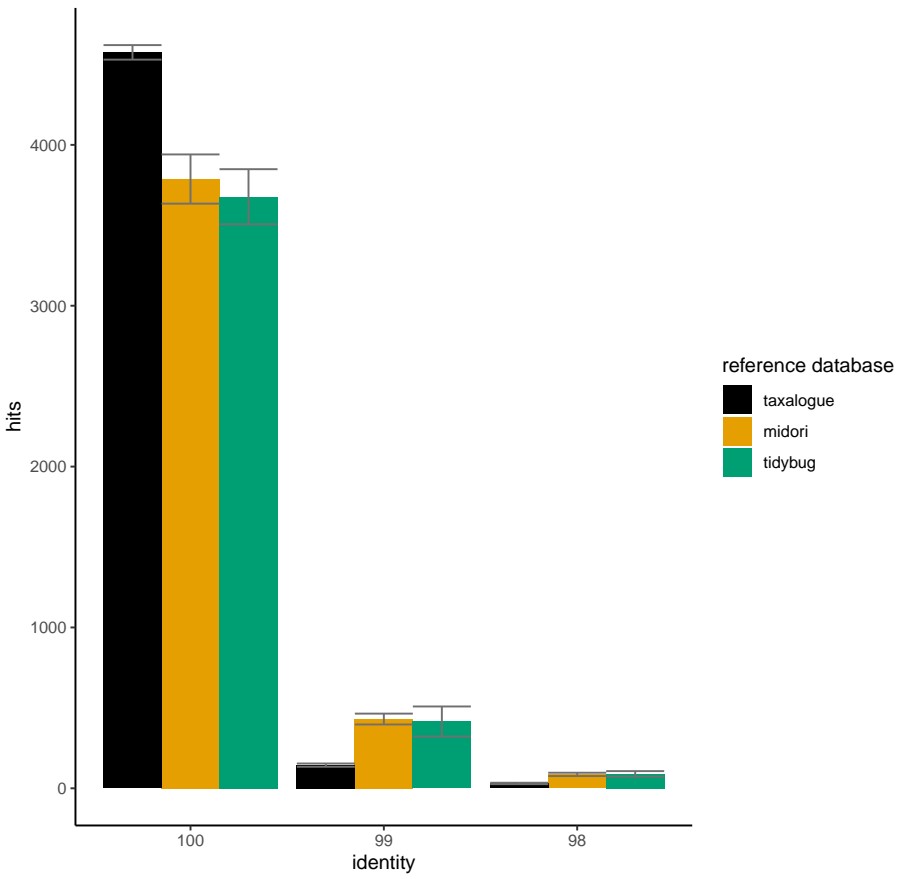

**Figure 4** **Top-hit identity distributions for the taxalogue, midori and tidybug reference databases queried against each other.** Each reference database was queried with 20 * 5,000 randomly selected sequences from all the other reference databases (*e.g.*, taxalogue was queried against sequences from midori and tidybug; midori was queried against sequences from taxalogue and tidybug, *etc.*). Only hits at 100, 99, and 98 percent identity were considered (see Table 1 for reference database descriptions). Each query consists of 5,000 randomly selected sequences. The whiskers show the standard deviation per reference database at a certain identity. Notes: identity = percent similarity between a query sequence and its best hit in a reference database; hits = number of best matches between query and reference database sequences at a certain identity percentage.

species level. Since (*Jin et al., 2020*) do not mention any measures to account for synonyms, the true error proportion might also be different. Furthermore, they identified a sequence as erroneous if the second-best hit (best hit would be itself) had a different taxonomic name, potentially leaving out other matches at 100% identity that could tag a record as erroneous.

Mock communities (samples with known compositions) could potentially be used to test the taxonomic assignment from differing reference databases. The results of the reference databases could be compared with the names of the mock community and subsequently summarized as a confusion matrix. However, this approach poses some problems. Since records from the reference databases are usually determined morphologically in the same way as records from the mock communities, the preference for one of these identifications

would be arbitrary. Furthermore, taxonomic mismatches between results are potentially due to synonyms or distinct taxonomic opinions of the taxon concept used. In our opinion, it is impossible to make an objective decision to accept the names of a mock community or the names of the reference databases as the truth. Therefore, we refrained from comparing the taxonomic assignments from the reference databases with those from a mock community.

As expected, merging records from commonly used source databases increases the coverage of a reference database (*Porter et al., 2014*; *Macher, Macher & Leese, 2017*; *Curry et al., 2018*; *Porter & Hajibabaei, 2018a*; *O'Rourke et al., 2020*; *Porter & Hajibabaei, 2020*; *Robeson et al., 2021*; *Nakazato & Jinbo, 2022*), at least for the reference database created by *taxalogue*. The porter reference database (*Porter & Hajibabaei, 2018b*), which also consists of records from GenBank and BOLD (see Table 1), on the other hand, has the lowest coverage of all tested queries. This is explainable because the last retrieval of GenBank records is from 2019, and it only uses the BOLD data releases, which are no longer updated since the end of 2015. The porter reference database also only uses records identified at species level, thereby discarding many records. This point illustrates that the source usage and the filtering of reference databases directly impact taxonomic coverage. However, an unexpected result is that the *taxalogue* reference database has a lower number of best hits at 100% identity with the Honduras query (see Fig. 2D). Since *taxalogue* downloaded records for all Arthropoda from BOLD, just as was done for tidybug. Still, tidybug has better coverage of the Honduras query, so either some records have been deleted from BOLD in the meantime, or *taxalogue* failed to download the respective records. After examining the missing sequences, we found that the missing sequences belonged to taxa with too many records in most cases. As mentioned in the Methods section, BOLD API downloads from taxa with numerous records are rarely successful due to read timeouts. *taxalogue* circumvents this problem by subdividing the failed taxon into lower taxa. Records only identified at the failed taxon level are unavailable since the BOLD API is subsequently queried with lower taxon levels (*e.g.*, from Arthropoda to Insecta, Arachnida, etc.). However, it is a relatively benign problem since the taxonomic resolution of those missing sequences is low (mostly family level and above). As this only occurs with record-rich taxa, we would expect a sufficient number of records with a high enough sequence identity to assign queries to, at least, the missing level. Due to the provision of release snapshots by BOLD, the download through API services is no longer required and using the releases with *taxalogue* (''classify --bold_release /path/to/release.tsv'') should be preferred. Using a release snapshot completely alleviates the aforementioned problems. Since the release processing requires more RAM and an account at BOLD, the download *via* the API will be maintained. This problem emphasizes the need for reproducible and transparent creation of reference databases. Since *taxalogue* logs the essential steps of the reference database generation, such issues are quickly resolved. Furthermore, it makes the creation of a reference database reproducible, which is indispensable for future replication or comparison.

## Reproducibility

Many BOLD records are private and not downloadable. Therefore, none of the reference databases tested do include private records. Solely consisting of downloadable records their coverage is of course reduced. For aquatic biota, up to 50% of sequences in some taxa were only available as private records (*Weigand et al., 2019*). The BOLD identification system allows comparing user-provided queries with private records, and included 9,458,738 records that could be used if private and public sequences were considered, but only 2,429,025 records were available if choosing only public sequences (accessed on the 4th of March 2022). Identification including private records increased the success rate from 43.3% to 78.6% for invasive pests, when using records from BOLD only (*Madden et al., 2019*). However, the usage of private records is flagged with a warning since the underlying database consists of unvalidated information (see also *Ratnasingham & Hebert, 2007*), and different taxonomic names for similar sequences might appear, especially from private records (*Ratnasingham & Hebert, 2013*). Furthermore, if the user compares the queries against all barcode records, no probability of placement is available. Another issue with this approach is that the records cannot be investigated and filtered based on meta-information or sequence quality. If a query has a hit with a private record, the user cannot investigate the sequence, which did cause problems in diagnosing pests (*Hodgetts et al., 2016*). And since the BOLD source database constantly changes, the taxonomic identification is not reproducible (*Federhen, 2011*).

For some private data, thorough reprocessing and curating misidentifications within the BOLD workbench might be the most important reason to delay a release (*Becker, Hanner & Steinke, 2011*). Additionally, the BOLD identification engine (as described in *Ratnasingham & Hebert, 2007*) remains largely a "black box" where the exact classification method is unknown. Several studies showed that classification methods varied in suitability on distinct reference databases compositions (*e.g.*, *Meier et al., 2006*; *Wilson et al., 2011*; *Virgilio et al., 2012*; *Bergsten et al., 2012*; *Lou & Golding, 2012*), so adjusting the classification method to the used reference database is crucial. In response to *Federhen (2011)*, BOLD added the option to identify a query against an annually created, time-stamped and archived reference database version (*Ratnasingham & Hebert, 2011*). These archived versions are a snapshot in time. They do not consider information deleted or changed over a year and therefore do not provide a reproducible identification if a user chooses the current version. The current version can change just within one day. Identification with a current version could consequently result in different outcomes within a single day. Since taxonomic name changes within BOLD are frequent (*Ratnasingham & Hebert, 2013*), a reproducible identification with private data could only be achieved if the identification was based on one of the archived versions of the reference database, and only if BOLD does not change the classification method. However, the usage of an archived version seems very unlikely since the latest version is from July 2019. BOLD did not add any versions since then. Archived versions are also not available for fungal or plant records. To preserve reproducibility and good scientific practice we refrained from adding any functionality that would incorporate private data. Nonetheless, software solutions providing this service have been developed (*e.g.*, https://github.com/VascoElbrecht/JAMP; *Yang et al., 2020*; *Buchner & Leese, 2020*).

## Geographic scale of reference databases

A reference database should be tailored to the needs of a particular research question (*e.g.*, *Mugnai et al., 2023*). Our case study compared global reference databases for Arthropoda, whereas a global scope might not be necessary for other research questions and could even hamper taxonomic identification (*Bergsten et al., 2012*). Using larger reference databases will certainly increase the number of erroneous sequences, although it is unclear if it increases the proportion of false positives. But using less comprehensive databases comes with the cost of potential false negatives. The main incentive to have an extensive database is to identify organisms at a higher taxonomic resolution with a more reliable identification (*Meyer & Paulay, 2005*; *Vences et al., 2005*; *Ekrem, Willassen & Stur, 2007*). Since a comprehensive database is also needed to distinguish closely related taxa with a great range (*Lou & Golding, 2012*; *Geiger et al., 2016b*), it is unknown at what point a local database might be the right choice to avoid the effect of decreased interspecific divergence of allopatrically distributed sister taxa in a geographically expanded dataset (*Bergsten et al., 2012*). Furthermore, the effects of geographical scale will differ between taxa and areas, and local reference databases could exclude invasive species or populations that have been recently shifting their ranges (*Bergsten et al., 2012*).

Which form of error is more acceptable has to be decided individually for each research question and could guide the reference database creation. To our knowledge, no current software is available with more extensive geographical filtering options than *taxalogue*. Reference databases could be filtered by multiple countries, continents, biogeographic realms, ecoregions, and even custom shapefiles. Geographic filtering reduces the effect of lower identification success due to a decreased genetic differentiation between closely related taxa in geographically broader reference databases (*Bergsten et al., 2012*). However, since online source databases hold records with missing location information (*Nilsson et al., 2006*; *Porter & Hajibabaei, 2018a*), or the available records are not evenly distributed across countries and continents (*Porter & Hajibabaei, 2018a*), geographical filtering has its limitations. Additionally, records rarely possess information about the coordinate reference system used–although most GPS trackers use the WGS84(EPSG:4326) by default.

## Taxonomic harmonization

Some data aggregators approximate a long-envisioned unitary taxonomy: a consensus classification and an entry point for additional taxonomic and nomenclatural information (*Thompson, 1993*; *Godfray, 2002*). *taxalogue* uses such unitary taxonomies to harmonize taxon names automatically. Harmonized taxon names are helpful due to the increasing usage of hierarchical classifiers in the taxonomic assignment step of a metabarcoding pipeline (*Piper et al., 2021*). Hierarchical classifiers depend on the taxonomic congruency between records since incongruent taxonomic information would introduce an artificial bias, leading to decreased identification success with lower taxonomic resolution. Other classification methods also benefit from harmonized reference databases because otherwise, a reference database could simultaneously consist of synonyms and the currently accepted name for one taxon, resulting in arbitrary assignments to the accepted or synonymized name. Taxonomic harmonization is already applied directly in NCBI and BOLD

(*Schoch et al., 2020*) and indirectly through the automated identification of specimens without prior taxonomic assignment (*Ratnasingham & Hebert, 2007*). To what extent users have harmonized identifications before uploading data to the source databases and on what basis is unknown. This indicates that taxonomic harmonizations occur to different and partly unknown degrees, even within a single source database. Data integration across multiple source databases, as in our test case, amplifies this problem since the records from different sources might also be harmonized to varying degrees. *Piper et al. (2021)* recommend taxonomic harmonization as a default step, just as other filtering procedures.

Even though taxonomic harmonization provides a clear advantage for further downstream analysis, criticism exists against synchronizing data to a particular unitary taxonomy. Such a taxonomy is algorithmically or socially resolved, even if no consensus has yet been reached in the taxonomist community (*Senderov et al., 2018*). A synthesized conclusion without clear consensus is suspected to decrease taxonomic stability (*Pauly, Hillis & Cannatella, 2009*) and trust in data aggregators (*Franz & Sterner, 2018*). Although macroecologists, conservationists, administrators and others depend on stable species lists for reliable predictions (*Hey et al., 2003*; *Isaac, Mallet & Mace, 2004*; *Padial & De la Riva, 2006*), the independence of taxonomy as a scientific endeavor has been stressed to be of utmost importance (*e.g.*, *Dubois, 1998*). A top-down administration is in stark contrast to taxonomic tradition (*Godfray, 2002*), where a taxon could be seen as a falsifiable scientific hypothesis that has to withstand time (*Haszprunar, 2011*). Scientists expressed concerns that such an administration would lead to authoritarianism (*Thiele & Yeates, 2002*) and about the data quality of biodiversity data aggregators (*e.g.*, *Franz & Sterner, 2018*). Even though we should preserve taxonomic independence, a non-taxonomist still has difficulties deciding which taxon name is most appropriate (*Grenié et al., 2022*). This problem is aggravated when very diverse taxa are studied or when different data sources are used (*Sterner & Franz, 2017*). Users should weigh the advantages of taxonomic harmonization against the disadvantages and decide accordingly.

*taxalogue* harmonizes with global backbone taxonomies, but regional or taxon-specific taxonomies may better represent the scientific consensus for that particular group. Since integrating many specialized taxonomies, with distinct scales and taxonomic breadth, is an enormous challenge and selecting appropriate taxonomies would still be opinion-based, we provide the commonly used NCBI Taxonomy, GBIF Taxonomy or no harmonization at all as options in *taxalogue*. However, we would like to point out that, for example, a specialized taxonomic harmonization, as found in *Arranz et al. (2020)*, might be a more appropriate choice for marine samples.

## Taxon concepts

Several studies showed that using taxon concepts *sensu* (*Berendsohn, 1995*) is necessary to unambiguously determine the meaning of a taxon name (*e.g.*, *Berendsohn, 1995*; *Kennedy, Kukla & Paterson, 2005*; *Franz, Peet & Weakley, 2008*). However, current source databases for sequence data do not provide this information. Of the major source databases, only BOLD provides a separate field for the used identification literature, which could help to derive the used taxon concept. Unfortunately, providing information for this field is

not an obligatory upload prerequisite. Furthermore, BOLD did not define this field's semantics. Therefore, it is unclear how to use this information. Consequently, reference database creation tools cannot provide taxon names in combination with the used taxon concepts. Instead, *taxalogue* approximates the idea of a reconciliation group (*Patterson et al., 2010*) with the option to generate a "comparison" file. This file aggregates previously used names for a taxon and aims to ease the information retrieval for all taxon names in the reference database. The taxonomic database Avibase is an example of how taxon concepts have already been implemented successfully (*Lepage, Vaidya & Guralnick, 2014*) and could guide further improvement of the source databases.

### Outlook

Since *taxalogue* combines sequences from up to three source databases, a user can achieve comprehensive coverage without relying on private and unreliable data, which posed problems in the past (*e.g.*, *Federhen, 2011*; *Hodgetts et al., 2016*). As a result, the reference database is reproducible and can be tailored to the particular research question. With comprehensive options to define the scale of the reference database, the user can exploit the advantages of a comprehensive (*Meyer & Paulay, 2005*; *Vences et al., 2005*; *Ekrem, Willassen & Stur, 2007*) and a local database (*Bergsten et al., 2012*) simultaneously. Furthermore, the options for taxonomic harmonization unlock the possibility of investigating their effects on the interpretation of taxa lists. The latter points to potential questions that future research still needs to address: To what extent do taxonomic harmonizations influence the significance of metabarcoding results? A harmonized and filtered reference database could also be a high-quality source for creating phylogenies. This is particularly interesting for combining gene-rich and species-rich data (*Chesters, 2017*), which could improve diversification analysis with phylogenies having higher and more accurate tree breadth (*Rainford, Hofreiter & Mayhew, 2016*). A further question is whether the absence of the taxon concept *sensu* (*Berendsohn, 1995*) in the source databases impedes the application of metabarcoding for ecological or macroevolutionary questions.

## ACKNOWLEDGEMENTS

We would like to thank Leighton Thomas for proofreading an earlier draft of the manuscript.

### Funding

This study was completed with financial support from the Leibniz Association (Leibniz competition grant K120/2018). The funders had no role in study design, data collection and analysis, decision to publish, or preparation of the manuscript.

### Grant Disclosures

The following grant information was disclosed by the authors:
Leibniz Association: K120/2018.

## Competing Interests

The authors declare there are no competing interests.

## Author Contributions

- Niklas W. Noll conceived and designed the experiments, performed the experiments, analyzed the data, prepared figures and/or tables, authored or reviewed drafts of the article, and approved the final draft.
- Christoph Scherber conceived and designed the experiments, authored or reviewed drafts of the article, and approved the final draft.
- Livia Schäffler conceived and designed the experiments, authored or reviewed drafts of the article, and approved the final draft.

## Data Availability

The data is available at Zenodo: Niklas W. Noll, Chirstoph Scherber, & Livia Schäffler. (2023). taxalogue: Associated data and code (Version 1) [Data set]. Zenodo. https://doi.org/10.5281/zenodo.6586571.

The code is available at GitHub and Zenodo:

- https://github.com/nwnoll/taxalogue
- Noll, N. W. (2023). taxalogue v0.9.3.3. Zenodo. https://doi.org/10.5281/zenodo.10014620.

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
