# Peer review of "taxalogue: a toolkit to create comprehensive CO1 reference databases"

_PeerJ, doi:10.7717/peerj.16253_

## Round 0.1 · original submission · Minor Revisions

The overall quality of the manuscript is good and I look forward to reading the revised version of the manuscript.

·

Basic reporting

In my opinion, not being a native English speaker, the manuscript is written in clear, unambiguous, and professional English language. The introduction and background sections appropriately describe the context, and the cited literature is appropriate, relevant, and well referenced. The structure of the paper conforms to PeerJ standards (besides the absence of a Conclusion section, but I’m not sure if it is mandatory or not) and the subsections used in the Discussion improve the clarity of the text. All the figures are relevant, high quality, well labelled and described. The source code of the developed software and all the code used for the validation are available in open-source repositories.

Experimental design

I think that this research, providing an original bioinformatic software tool, is within the scope of the journal. The authors clearly state the knowledge gap they are trying to fill with their research. They are also aware of all the pros and cons of their approach which are clearly explained and contextualized within the actual knowledge in this field. All the methods have high technical levels and are described with sufficient details to allow reproducibility.

Validity of the findings

The rationale and benefits of the findings of this research are clearly stated. All underlying data are statistically sound and have been provided.

Additional comments

MAJOR COMMENTS:
I tried to use to use taxalogue and it generally worked well, but I encountered an issue with the GenBank database. In fact, I was not able to download anything from this database, I tried with different taxon (also with single genus/species with few sequences) but it failed every times. I had no time to verify if the problem persists using different operating systems (I used ubuntu 18.04 on a wsl for windows) or ruby versions (I used ruby 3.0.5), but I strongly suggest the authors to double check if this issue could affect other users.

MINOR COMMENTS:
The methods used are generally very well described, but I think that the section on the case studies, especially the part on the comparison of the ranked data and the queries with the subsampled data (lines 278-291) could be expanded (e.g., describing the methods used for ranking) and clarified (e.g., using less technicisms) a bit to make it more comprehensible for readers not working in this field.
The references of the datasets used for the validation are cited in the table 2 but they are missing from the references section, I suggest adding them to this section. Moreover, I really appreciate that the authors provided all the codes and supplementary data in a public repository (zenodo) but, even if some of these data are mentioned in the text (lines 264-265), the url for the repository /https://doi.org/10.5281/zenodo.6586571) is not reported, please add it.

Reviewer 2 ·

Basic reporting

This study developed software for creating the reference database, which is useful for future reference by other researchers. However, there are several issues that need to be addressed. All issues are listed below.

Experimental design

The problem statements and the aim of this study are well-stated.

Validity of the findings

The authors have provided the case study and compared it with other databases to demonstrate the reliability of the software.

Additional comments

-It is better to add a section on the implementation of the software, even though it is stated in github.
-DNA Barcode or DNA barcode, folmer or Folmer. Be consistent.
-The figure citation for Fig. 2 – 4 is better to be cited in the results section only.
-All abbreviations should be introduced, e.g., CO1, BOLD, GBIF, etc.
-Line 79: Double-check the sentence.
-Line 177: Avoid using won’t.
-Suggestion: It is better to add URL for the reference databases in Table 1.

·

Basic reporting

The handling of the language is adequate. I made some suggestions as to how it seems more understandable to the public that it is unrelated to the area.

The literature is adequate. I suggested a little more in terms of taxonomic harmonization.

The color palette is a little hard to follow. From my point of view the black color should be avoided but it is only a suggestion.

The comparison with other systems is fair but not exhaustive

Experimental design

The design of the software seems good to me and includes the deficiencies in other similar software.
The way it is built will allow for long-term development and not just a short-term solution.
The installation methods and the user manual are clear and allow reproducibility

Validity of the findings

The installation of the software from the provided guide is not so immediate.
I had problems with the version of
# gem install bundler:2.2.3
Especially if you work on a computer cluster without root access.

In general, after the complications in the installation, the software works and the example in the article can be replicated.

Additional comments

It seems to me that the tool is quite promising and has a lot of potential.

It is a bit complicated that it can be easily integrated with other more user-friendly software ecosystems (such as R) however it is not that complicated to use either because the manual is well built.
Installation can be a bit of a problem if you don't have enough console expertise.
I suggest if possible a smoother installation with a self-contained script. Above all the problem is that some gems collide with others if you use a more recent version of the rails and it also gets complicated if you don't have root access in a cluster.

I also suggest that you identify yourself with your entrez API credentials for NCBI as this can considerably improve download times and volumes

---

## Round 0.2 · Minor Revisions

I have only minor concerns regarding language formatting errors (only some are highlighted below). I would recommend the authors kindly proofread the whole manuscript again for any possible discrepancies.

Line 15: why is "Metabarcoding" in capital letter?
line 22-23: "In combination with extensive geographical options..." The phrase is repeated twice.
Line 46 (and others): why is 'Barcode' in capital letter>?
line 354: there is a space after 'tested'.
Line 406-407: "This leads to the problem that records only identified until the failed taxon level IS unavailable.". Grammar error. Additionally, I do not comprehend what the authors are trying to portray. Do you mean the after taxalogue subdivides the failed taxon into lower taxa, thereby circumventing the problem of BOLD API downloads, it leads to another problem?

---

## Round 0.3 · accepted · Accept

Thank you for following through with my comments. I think the manuscript is now ready for publication.